# Solid-Liquid Triboelectric Nanogenerator Based on Vortex-Induced Resonance

**DOI:** 10.3390/nano13061036

**Published:** 2023-03-13

**Authors:** Xiaowei Li, Di Zhang, Dan Zhang, Zhongjie Li, Hao Wu, Yuan Zhou, Biao Wang, Hengyu Guo, Yan Peng

**Affiliations:** 1School of Mechatronic Engineering and Automation, Shanghai University, Shanghai 200444, China; 2Institute of Artificial Intelligence, Shanghai University, Shanghai 200444, China; 3Department of Applied Physics, Chongqing University, Chongqing 400044, China; 4Shanghai Artificial Intelligence Laboratory, Shanghai 200232, China

**Keywords:** vortex-induced vibration, solid-liquid triboelectric nanogenerator, computational fluid dynamic

## Abstract

Energy converters based on vortex-induced vibrations (VIV) have shown great potential for harvesting energy from low-velocity flows, which constitute a significant portion of ocean energy. However, solid-solid triboelectric nanogenerators (TENG) are not wear-resistant in corrosive environments. Therefore, to effectively harvest ocean energy over the long term, a novel solid-liquid triboelectric nanogenerator based on vortex-induced resonance (VIV-SL-TENG) is presented. The energy is harvested through the resonance between VIV of a cylinder and the relative motions of solid-liquid friction pairs inside the cylinder. The factors that affect the output performance of the system, including the liquid mass ratio and the deflection angle of the friction plates, are studied and optimized by establishing mathematical models and conducting computational fluid dynamics simulations. Furthermore, an experimental platform for the VIV-SL-TENG system is constructed to test and validate the performance of the harvester under different conditions. The experiments demonstrate that the energy harvester can successfully convert VIV energy into electrical energy and reach maximum output voltage in the resonance state. As a new type of energy harvester, the presented design shows a promising potential in the field of ‘blue energy’ harvesting.

## 1. Introduction

Recent decades have witnessed rapid development in integrated circuits, and MEMS technology has made electronic devices much smaller in size and considerably more energy-efficient. However, these electronic devices still rely on batteries for power supply, and the safe disposal of chemical waste poses a particular challenge [1,2]. Therefore, harvesting energy from the surrounding circumstance is a more reasonable solution. Blue energy contains tidal energy, ocean current energy, temperature difference energy, etc. [3,4]. Among them, ocean current energy is a widely recognized source of clean energy. The main method of harnessing ocean current energy is to utilize turbines, which work better in areas with high flow velocities. However, turbines struggle to function efficiently in low-flow speed environments, whereas vortex-induced vibration can produce periodic oscillations, and the vibration energy can effectively convert low-speed flow energy into electrical energy.

Vortex-induced vibration is a typical fluid-structure interaction phenomenon caused by the viscous effect of fluids, which generates alternating shedding vortices around the back of the spoiler cylinder and causes the cylinder to reciprocate motion perpendicular to the flow direction. Moreover, the structure can still vibrate, even at a low flow speeds. In the early days, research on vortex-induced vibration mainly focused on prevention and suppression to prevent damage to the structure. M.L. Facchinetti et al. [5] simulated a low-order model of vortex-induced vibration with three coupling terms and found that the acceleration coupling model performed better. Bearman [6] elaborated detailed conclusions on the response characteristics of rigid cylinders to one degree-of-freedom (1 DOF) VIV. Moe and Wu [7] conducted forced and free vibration experiments on the cylinder, and the results can predict the amplitude of a cylinder in a given flow. Zhou et al. [8] studied the elastic cylinders via the discrete vortex method incorporating the vortex-in-cell (VIC) and found the maximum amplitude of vibration can reach 0.57 diameter. Bishop and Hasson [9] expounded the experimental conclusions of lift and drag forces on the vibrating cylinder and proposed a wake oscillator model. Parkinson [10] qualitatively described major features of near-wake vortex shedding and VIV using analytical and numerical methods. 

From another perspective, the VIV energy can convert kinetic energy into mechanical energy, which can be recovered and utilized in low-velocity water flow through energy harvesters. At present, many researchers have carried out related research on vortex-induced vibration energy harvesting. Junlei Wang et al. [11] proposed a bluff body with different interfaces that can couple VIV and gallop to improve energy harvesting performance. Iman Mehdipour et al. [12] conducted a comprehensive experimental study on the shape of a blunt body causing VIV and enhance the performance of low-speed wind energy harvester. Toma et al. [13] proposed a vertical moving cylinder piezoelectric energy harvesting device. Baoshou Zhang et al. [14] proposed a novel biomimetic V-shaped layout, increasing the energy recovery area of downstream cylinders. Ying Gong et al. [15] proposed a VIV energy harvester with a direction-adaptive wing, which is well adapted to the direction-varying flow environments. Zhaoyong Mao et al. [16] studied the effect of spacing between cylinders on VIV energy harvesting and optimized the layout of four cylinders. Junlei Wang et al. [17] introduced metasurface into VIV and found that certain surfaces can increase voltage and amplitude. Mingjie Zhang et al. [18] studied the effect of the Reynolds number on energy harvesting of cylindrical VIV through piezoelectric energy harvesters. Mengfan Gu et al. [19] studied the effect of submerged depth on VIV energy harvesting and found that the conversion efficiency reached a peak at 0.5 m. Yufei Mei et al. [20] greatly improved the net energy output of VIV by controlling jet flow through deep reinforcement learning. Peng Han et al. [21] studied the vibration and energy harvesting characteristics of low-mass square bluff bodies at different incident angles. 

Energy harvesters that convert mechanical energy into electrical energy are predominantly divided into three categories according to the principle: electromagnetic, piezoelectric, and electrostatic [22,23]. Among them, the electromagnetic energy harvesters are large in size and high in cost, and the piezoelectric energy harvesters can only be effective under high-frequency vibrations [24,25]. As an emerging energy harvesting technology, TENG utilizes triboelectric electrification and electrostatic induction to convert mechanical energy into electrical energy. It exhibits the advantages of low cost, light weight, and, particularly for low-frequency vibration energy harvesting, presents better performance [26,27]. Two main types of TENG have been invented, namely, solid-solid TENGs and solid-liquid TENGs. 

Recent decades have witnessed various prototypes and structures of solid–solid TENGs. Wang et al. [28] were the first to demonstrate the conversion of mechanical energy into electrical energy via triboelectric electrification and electrostatic inductive coupling. As the research progressed, nanoscale triboelectric-effect-enabled energy conversion was discovered, which was named the triboelectric nanogenerator due to the nano-scale surface roughness. Zize Liu et al. [29] proposed a temperature sensor by TENG to detect human body temperature without an external power supply. Jing Liu et al. [30] proposed a triboelectric hydrophone for monitoring underwater low-frequency sound signals and offering high-sensitivity. Yan Wu et al. [31] fabricated a generator that combines TENG and an electromagnetic generator (EMG) to collect low-frequency wave energy. Liang Xu et al. [32] studied different connection methods of the TENG network and improved the performance of TENG cluster. Huamie Wang et al. [33] improved the performance of TENGs by encapsulating the shuttle of charges in conductive domains and increasing the charge density. Tian Xiaoxiao et al. [34] used silicone rubber and carbon black as flexible electrodes, improving the performance of TENG. Bolang Cheng et al. [35] proposed a temperature differential TENG that can improve the performance in high-temperature circumstances. Nonetheless, the solid-solid TENGs faced several problems: first, the wear of the solid-solid friction pair diminishes the long-term working stability of TENGs. Additionally, high humidity normally exerts an adverse effect on the performance of solid–solid TENGs, while solid-liquid TENGs have drawn much attention for their stable output performance and durability.

Xiaoyi Li et al. [36] proposed a solid-liquid TENG of nanowires, which significantly increases the contact surface of fluorinated ethylene propylene (FEP) with water and improved the performance. Lun Pan et al. [37] designed a U-tube SL-TENG to study the effect of liquid properties on output performance. Wei Tang et al. [38] developed a mercury-based triboelectric nanogenerator (LM-TENG) with significantly improved output efficiency. Xiya Yang et al. [39] developed a polytetrafluoroethylene (PTFE)-based triboelectric nanogenerator for water wave energy harvesting. Jinhui Nie et al. [40] proposed a triboelectric nanogenerator based on the interaction of two pure liquids. Shunmin Zhu et al. [41] developed a triboelectric nanogenerator without solid moving parts for waste heat recovery. Jing You et al. [42] proposed the equivalent circuit model of solid-liquid TENG and solved the electrical phenomena on the liquid-solid interface. Cun Xin Lu et al. [43] studied the effect of ambient temperature on the output performance of triboelectric nanogenerators. Liqiang Zhang et al. [44] found that temperature, ionic concentration, and pH affected the output of SL-TENGs via experimental research. Liyun Ma et al. [45] developed smart protective clothing based on triboelectric nanogenerators, which collect motion to power the monitoring system. Xinkai Xie et al. [46] utilized TENG to design a self-powered gyroscope angle sensor to detect the relative rotation angle. Song Wang et al. [47] designed a self-powered tilt sensor based on an SL-TENG, which demonstrates stable output and high sensitivity. Due to the stability and insensitivity to humidity, SL-TENGs are more suitable for ocean flow energy harvesting.

In recent decades, numerous researchers have systematically studied VIV and summarized its characteristics. At the same time, many researchers have proposed energy harvesters based on VIV and optimized the power generation efficiency of the energy harvester via various methods. With the proposed triboelectric nanogenerators, this technology has been applied in many scenarios in recent years. However, no research on the SL-TENG was utilized to collect VIV energy. 

In this paper, a VIV-SL-TENG is proposed for harvesting low-velocity flow energy in oceans and rivers. The SL-TENG is embedded in a cylinder to collect vortex-induced vibration energy. As shown in Figure 1, when the device is placed in a water environment, it generates vibrations under the action of water, and the internal SL-TENG unit converts the vibration energy into electric energy. More details are shown below. The main novelties are as follows. Firstly, we establish a theoretical model to explore the physical parameters that influence the state of the cylinder. In addition, we investigate how the cylinder diameter and the speed affect the vortex frequency by computational fluid dynamics (CFD). Then, we find that the VIV-SL-TENG performs best in the resonance state. At the same time, in the resonance state, to increase the efficiency of the internal liquid-solid friction pair, we investigate the performance of the VIV-SL-TENG under different deflection angles.

## 2. Materials and Methods

### 2.1. Experiment Platform and TENG Working Mechanism

We built an experimental platform, as shown in Figure 2a; the VIV-SL-TENG is vertically installed in a flume that provides a continuous and stable water flow. Vortex-induced vibration occurs on the VIV-SL-TENG under the impact of water flow, which converts the kinetic energy of the water flow into vibration energy, making it possible to capture the water flow energy. The VIV-SL-TENG can be installed in situations where water flows, such as rivers and oceans, to continuously power devices. A schematic diagram of a single VIV-SL-TENG is shown in Figure 2b; the VIV-SL-TENG is primarily composed of a cantilever beam, a hollow cylinder, a guiding disc, a carrier sheet, and a solid-liquid triboelectric nanogenerator unit. The SL-TENG is fastened to the carrier sheet, and the carrier sheet is assembled on the guide disc. Sequentially, the guide disc is mounted inside the cylinder. As shown in Figure 2c, a single electrode TENG was prepared using an FEP film as the dielectric film. The TENG consists of a dielectric material and an electrode (copper). A copper film (a thickness of 125 µm) is adhered between two FEP films. The area of the FEP films (30 mm × 60 mm, the thickness of FEP films is 50 µm) is larger than that of the copper film (20 mm × 40 mm) to ensure that the copper film is entirely isolated from water. In addition, we utilized deionized water as the liquid triboelectric material to construct the entire TENG. To achieve resonance, a finite element analysis was conducted for the structure, which yielded a natural frequency of 2 Hz. The Strouhal number (St) was then determined based on the flow velocity, which was limited to 1 m/s. Then the vortex shedding frequency could be calculated according to the flow velocity and the model parameters, as shown in Table 1.

As shown in Figure 2a, the cylinder is entirely immersed in the water of the flume. The entire VIV-SL-TENG reciprocates under the action of VIV. The deionized water inside the cylinder moves along with the VIV-SL-TENG, causing the relative motion between the contact of the deionized water and the FEP. The working mechanism of the solid-liquid TENG is shown in Figure 2d; in the initial state (Figure 2d(I)), we assume that the surface of the dielectric material and water is electrically neutral. When the deionized water shakes along with the VIV-SL-TENG, electrons move from the deionized water to the surface of the FEP due to the different abilities to attract electrons. This results in the surface of the FEP becoming negatively charged and the water interface becoming positively charged, as shown in Figure 2d(II). When the VIV-SL-TENG moves to the limit position, the contact area between the deionized water and the FEP reaches a maximum, and a large number of charges accumulate on the FEP surface (Figure 2d(III)). Subsequently, the deionized water begins to move away from the FEP surface, and the charges on the contact surface are separated. This causes transferred electrons to remain on the FEP, resulting in the potential of the deionized water to be higher than that of the induced electrode. Therefore, a potential difference is generated between the induced electrode and the ground, driving current from the ground to the electrode (Figure 2d(IV)). During the motion of the deionized water, electrons flow continuously until the charge on the FEP reaches saturation (Figure 2d(V,VI)). When the cylinder and the deionized water reach the other limit position, the deionized water flows towards the FEP again (Figure 2d(VII)). At this point, due to the gradual decrease in the potential difference with the increase in contact area, the current flows back to the ground, as shown in Figure 2d(VIII). Then the electrons continue to flow until the deionized water moves to the limit position again (Figure 2d(IX)), and then the deionized water returns to the initial state and starts the next cycle (Figure 2d(X)).

### 2.2. Mathematical Model

We consider the cylinder as a one degree-of-freedom elastic-supported rigid cylinder with a diameter of *D*. The VIV phenomenon is a non-linear process that involves self-excitation, and the VIV can cause the system to exhibit large-amplitude oscillations. A van der Pol oscillator, as a non-linear process, can better capture the dynamics of the system and can be a useful representation of the self-excited behavior seen in VIV. Therefore, a novel van der Pol model was adopted to establish the kinematic coupling equation between the rigid cylinder. The vibration response of the cylinder is obtained by decoupling. As shown in Figure 3, *x* is the down-flow direction, *y* is the vibration direction, and *U* is the flow speed. We assume that the cylinder only moves in one direction, namely, the lateral displacement *Y* in the *y* direction. The vibration equation can be written as follows:(1)$$m\ddot{Y}+r\dot{Y}+hY=S.$$
where (·) is the derivative concerning the dimensional time *T*, *S* is the forcing caused only by vorticity in the wake, and the mass *m* includes the sum of the rigid cylinder mass *m_s_* and the additional mass *m_f_* of the fluid.
(2)$$m=m_{s}+m_{f},\;m_{f}=C_{M}\rho^{2}\beta \pi /4,\;\mu =\frac{m_{s}+m_{f}}{\rho D^{2}},\;S=\frac{1}{2}\rho U^{2}DC_{L}$$
where *ρ* is the density of water, *μ* is the mass ratio, and *C_M_* is the additional mass coefficient. In Equation (1), *r* is damping, including structural damping *r_n_* and fluid resistance *r_f_*.
(3)$$r=r_{n}+r_{f},\;r_{f}=\gamma \Omega \rho D^{2},\;\Omega =2\pi StU/D.$$
where *γ* is a stall parameter, Ω is the angular frequency, and *St* is the Strouhal number.

In this paper, only the low-order coupling effect of transverse VIV on the wake vibrator was considered, and only the transverse vibration displacement generated was calculated. Therefore, a 1 DOF transverse vibration wake oscillator model was established based on the work of Facchinetti et al. Facchinetti discussed the coupling effects of displacement, velocity, and acceleration on the wake oscillators and concluded that the form of acceleration coupling better reflects the vortex-induced vibration. Therefore, in this article, we also chose the transverse acceleration to represent the forcing term. The vibration characteristics of the vortex street were simulated by a nonlinear oscillator satisfying the van der Pol equation:(4)$$\ddot{q}+\varepsilon \Omega_{f}(q^{2}-1)\dot{q}+\Omega_{f}^{2}q=F.$$
(5)$$\Omega_{f}=2\pi \frac{StU}{D}.$$

By dimensionless normalization of the cylinder vibration equation and wake vibrator equation, the dimensionless time $t=T\cdot \Omega_{f}$, y=YD is introduced, respectively, and the two dimensionless quantities are substituted into Equations (1) and (4):(6)$$\begin{array}{l} \ddot{y}+(2\xi \delta +\frac{\gamma }{\mu })\dot{y}+\delta^{2}y=Mq\\ \ddot{q}+\varepsilon (q^{2}-1)q+\dot{q}=A\ddot{y}. \end{array}$$
where *ξ* is structure-reduced damping, *M* is essentially a mass number and scales the effect of the wake on the structure, *A* is the coupling force coefficient of the structure on the fluid, and *ε* is a small parameter in a nonlinear term, and among them,
(7)$$M=\frac{C_{L0}}{16\pi^{2}St^{2}\mu },\;\delta =\frac{1}{StU_{r}},\;U_{r}=\frac{2\pi U}{\Omega_{S}D}.$$

The amplitude of the linear transfer equation between the displacement of the cylinder and the variable of fluid is
(8)$$\frac{y_{0}}{q_{0}}=M{[(\delta^{2}-\omega^{2}{)}^{2}+{\left(2\delta \xi +\frac{\gamma }{\mu }\right)}^{2}\omega^{2}]}^{-0.5}.$$

Then, substituting in the wake oscillator equation and only considering the major harmonic contributions of nonlinearity, the equation for the amplitude *q*_0_ and frequency ratio *ω* can be obtained via elementary algebra:(9)$$\omega^{6}-[1+2\delta^{2}-{\left(2\xi \delta +\gamma /\mu \right)}^{2}-AM]\omega^{4}-[-2\delta^{2}+{\left(2\xi \delta +\gamma /\mu \right)}^{2}-\delta^{4}+AM\delta^{2}]\omega^{2}-\delta^{4}=0.$$
(10)$$q_{0}=2{\left[1+\frac{AM}{\varepsilon }\frac{(2\xi \delta +\gamma /\mu )\omega^{2}}{{\left(\delta^{2}-\omega^{2}\right)}^{2}+{\left(2\xi \delta +\gamma /\mu \right)}^{2}\omega^{2}}\right]}^{0.5}.$$

The values of *ω*, *y*_0_, and *q*_0_ can be obtained via Equations (8)–(10), respectively, representing the frequency ratio, dimensionless maximum amplitude, and lift oscillator amplitude, respectively.

As shown in Figure 4a–c, they represent the variation tendencies of the cylinder motion amplitude, lift oscillator, and frequency ratio at different flow velocities when *μ* = 0.93 and *ξ* = 0.03. As shown in Figure 4a, when the flow speed increases, the frequency ratio decreases slowly and remains at about 1, indicating that the vibration frequency of the cylinder is close to its natural frequency. As shown in Figure 4b, the lift oscillator first increases and then decreases with the increase of flow speed, reaching the maximum when *U_r_* = 5 because resonance occurs at *U_r_* = 5, according to the design and calculation. When *U_r_* is less than 5, the lift oscillator rises rapidly and then drops slowly. As shown in Figure 4c, the variation trend of the maximum displacement vibration amplitude along with the flow speed is the same as that of the lift oscillator. Before the locked region, when the flow speed increases, the vibration amplitude of the rigid cylinder, namely, *y*_0_, increases significantly, and the lift oscillator also gradually increases, reaching the maximum in the locked region. The locked region is around *U_r_* = 4.5–6, and both before and after this interval are non-locked phases. When *U_r_* is larger than 6, the amplitude slowly decreases with the increase of flow speed. However, the vibration amplitude reaches the maximum when *U_r_* = 6 instead of *U_r_* = 5. As a result, the displacement amplitude caused by locking increases to the maximum point rather than at the frequency ratio of 1, which reflects the phenomenon of mistuning during locking. At the same time, when the vortex shedding frequency is close to the cylinder vibration frequency, the structure is locked.

Next, we studied the VIV response characteristics of a rigid cylinder. The main parameters that affect the VIV of a rigid cylinder are the flow speed, mass ratio, and structure damping ratio. Here, we discussed the influence of the mass ratio *μ* and the structural damping ratio *ξ* on the maximum amplitude, lift force, and frequency ratio. Figure 4d–f describe how the mass ratio affects the frequency ratio, maximum amplitude, and lift amplitude. As the mass ratio increases, the lift amplitude and the maximum amplitude decrease, and the range of the locked region also decreases. Then, Figure 4g–i describe how the structural damping ratio affects the maximum amplitude, lift amplitude, and frequency ratio, respectively. They show that the maximum displacement response decreases significantly when the structural damping ratio increases from 0.03 to 0.1, and the locked region is steady, and the influence on lift decreases with the increase of the structural damping ratio. By comparison, the mass ratio presents a significant impact on the VIV characteristics, while the structural damping presents a weaker impact.

### 2.3. Simulation

To determine the basic characteristics of the cylinder from the water flow, 2D transient numerical simulations were employed to qualitatively analyze the causes for the induced flow-induced force. As an effective approach, researchers have taken advantage of CFD in VIV research in recent years. In this article, we utilized the 2D Reynolds-Averaged Navier–Stokes (RANS) equations accompanied by the *k*-*ω* SST turbulence model to simulate the flow field. As shown in Figure 5a, the diameter *D* of the cylinder is 0.05 m, and the computational domain is 10 *D* × 30 *D*. The distance between the cylinder and the speed inlet is 10 *D*, while the distance between the cylinder and the pressure outlet is 20 *D*. The vortex contour of the cylinder at 0.5 m/s is presented in Figure 5b, which shows a series of vortices appearing behind the cylinder. 

The lift force and amplitude of the cylinder at 0.5 m/s are shown in Figure 5c,d. The lift force of the cylinder fluctuates with time, the same as amplitude, and both lift and amplitude show periodicity clearly. The frequency of the lift force can be calculated by fluorinated ethylene propylene (FEP), as shown in Figure 5e; the frequency is 2 Hz, which is equal to the vortex shedding frequency calculated by the equation above. In addition, the natural frequency of the harvester is also 2 Hz. Therefore, after design and calculation, the harvester will reach resonance at 0.5 m/s.

To better understand the basic characteristics of VIV in water flow, we investigated the effects of speed and diameter on the lift force, amplitude, and frequency. When the diameter *D* = 50 mm, the effect of speed is shown in Figure 5f–h, with a flow speed range of 0.35–0.7 m/s. As shown in Figure 5f, the lift force increases slowly with the increase of flow speed. However, as shown in Figure 5g, the amplitude initially increases and then decreases with the flow speed and reaches the maximum value at 0.5 m/s. Figure 5h shows that the frequency is not sensitive to the speed, which is consistent with the above theoretical calculation results. When the flow speed *U* = 0.5 m/s, the effect of diameter is shown in Figure 5i,j, with a diameter *D* the range of 3–7 cm. And both the lift and the amplitude increase with the increase of diameter. Figure 5k shows that the frequency is stable at about 2 Hz, which is close to the natural frequency.

To investigate the influential factors of the SL-TENG, we simulated the potential distribution of the TENG by COMSOL Multiphysics software. In this approach, we neglected the influence of fluid speed to facilitate the simulation. Then, we chose ethanol (Figure 6a), deionized water (Figure 6b), and gallium (Figure 6c) as liquid triboelectric materials to investigate the effects of liquids on potential distribution. Furthermore, as shown in Figure 6d–f, we investigated the effects of the contact area between the deionized water and the FEP on potential distribution.

As shown in Figure 6a–c, the potential distribution of SL-TENGs with different kinds of liquids demonstrated that the output performance of gallium was higher than those of the other two kinds of liquid. Subsequently, Figure 6d–f show that as the contact area between the deionized water and the FEP increased, the potential difference also increased. Thus, we assume that a larger contact area means more transferred charges.

## 3. Results and Discussion

### 3.1. Measurement and Characterization

As shown in Figure 7a, an experimental platform was set up on the upper part of the flume. The flume could provide continuous and stable water flow, and the flow speed was measured by a doppler current meter (P 25703), and the current speed monitoring is shown in Figure 7b. For the electrical signal of TENG, we employed an oscilloscope (Tektronix MDO3024) to measure the voltage. The two electrodes of the oscilloscope were connected to the ground and the copper electrode of TENG, respectively.

### 3.2. Experimental Results

To illustrate the influence of the flow speed on VIV and the output performance of the SL-TENG, we conducted experiments under the same initial liquid level height and deflection angle, and we only adjusted the valve to control the flow speed of water in the flume. The energy harvester was fixed in the middle of the flume so that the cylinder part was completely submerged in the water, and the flow range was set to 0.35–0.75 m/s. Figure 8a–c show the snapshots of the VIV-SL-TENG at 0.4 m/s, 0.5 m/s, and 0.6 m/s, respectively. Figure 8b shows the cylinder presented a significant amplitude at 0.5 m/s. Furthermore, a series of vortices could also be distinctly seen behind the cylinder (Appendix A).

Figure 8d shows the voltage signal of the VIV-SL-TENG when the flow speed was 0.5 m/s. Figure 8d shows that the voltage signal was stable, uniform, and periodic, with two peaks occurring within one second. This periodicity is in good agreement with the simulation of VIV above, indicating that the voltage was generated from the VIV. In addition, the voltage signal was constantly accompanied by noise, which was found to be caused by friction between the outer wall of the energy harvester and the flowing water after the experimental investigation. As shown in Figure 8e, the open-circuit voltage of VIV-SL-TENG increased first and then decreased with the increase of the water flow speed, reaching the maximum value of 1.48 V at about 0.5 m/s, and decreasing to 1.12 V at 0.75 m/s. This trend is consistent with the theoretical calculations presented above, as both lift and amplitude reached their maximum at 0.5 m/s. Furthermore, according to the calculation, the VIV-SL-TENG reached the resonance state at 0.5 m/s, where the amplitude reached the maximum. At this time, the contact area between the deionized water and the friction plate was the maximum, and thus the voltage reaching the maximum value. When the flow speed was faster or slower than 0.5 m/s, the energy harvester could not achieve the maximum amplitude state, and thus the voltage produced was less than that at 0.5 m/s.

To maximize the efficiency of the TENG, it is necessary to increase the contact area and the contact speed between the deionized water and FEP as much as possible. To explore the factors affecting the motion state of the deionized water in the cylinder, we carried out simulation analysis by STAR-CCM+. For different initial liquid level heights, as shown in Figure 8f, with the increase of the initial liquid level height, the peak-to-peak value initially increased and then decreased, and reaching the maximum at 3 cm. In addition, we conducted an experimental study on the influence of the liquid level height. The installation position of the energy harvester was consistent, and the water flow speed was maintained at 0.5 m/s. The influence of the initial liquid level height in the range of 2–5 cm on the voltage output was studied. As shown in Figure 8f, the voltage response also initially increased and then decreased, and reached the maximum output at 3 cm, with a maximum voltage of 1.4 V and a minimum voltage at 5 cm, with a voltage of 1.32 V. The trend of voltage is consistent with the peak value of the liquid level, reaching the maximum value at 3 cm, which once again verifies the influence of contact area on voltage output. 

Then, we executed simulations and experimental research on the influence of the deflection direction of the friction plate inside VIV-SL-TENG. First, we simulated the motion state of the liquid inside the cylinder by STAR-CCM+. After preliminary analysis, we varied the deflection angle from 0° to 360°, with an interval of 36°. The influence of the deflection angle on the peak-to-peak value was explored through 3D CFD simulation, with the friction plate positioned at the edge of the cylinder. As shown in Figure 9b, when the vibration direction was perpendicular to the friction plate, the peak-to-peak value reached the maximum, while when the friction plate was parallel to the vibration direction, the peak-to-peak value reached the minimum.

In addition, we conducted experiments to explore the effect of deflection angle on the output voltage. From the previous experiments, the VIV-SL-TENG achieved the highest voltage output at a flow speed of 0.5 m/s. Therefore, the deflection angle experiment was set at *U* = 0.5 m/s, with the initial height of the liquid level was 3 cm, and the friction plate was deflected by 36° each time. As shown in Figure 9b, the vibration direction of the energy harvester was perpendicular to the water flow direction. When the direction of the friction plate was perpendicular to the vibration direction, the voltage reached the maximum, namely, 1.44 V. The voltage gradually decreased with the deflection angle increases until the friction plate was parallel to the vibration direction. Hence, the deflection angle presented a significant impact on the voltage output of the SL-TENG. The peak-to-peak value of the liquid level obtained by the simulation was consistent with the voltage output, indicating that the voltage output of the energy harvester was closely related to the peak-to-peak value of the liquid level. In addition, we carried out a charging experiment under 0.5 m/s, as shown in Figure 9a; we charged a smaller capacitor, which was 22 μF. The corresponding voltage stabilized and reached the maximum at 120 s, with a voltage of 147 mV. In summary, the experimental results indicated that the VIV-SL-TENG performed well in the vortex-induced resonance. And further work will focus on optimizing the system and exploring the influence of the shape of the bluff body on VIV-SL-TENG.

## 4. Conclusions

In summary, we proposed a novel VIV-SL-TENG. Furthermore, we simulated and analyzed the lift and amplitude of the cylinder under the impact of the water flow via the CFD method. Subsequently, we established a mathematical model to calculate the responses of VIV and discussed the influence factors of VIV through a parametric study. Based on theoretical and simulation analysis, we prototyped the concept and conducted extensive experiments with different flow velocities and deflection angles. The experimental results indicated that the harvester performed well in the resonance state. The voltage output reached a maximum of 1.48 V at a flow velocity of 0.5 m/s. Furthermore, we investigated the influence of initial liquid level height and deflection angle by experiments, and the voltage output reached the peak at approximately 3 cm, and the deflection angle was positively correlated with the peak-to-peak value of the liquid level. The voltage reached the maximum when the direction of the friction plate was perpendicular to the vibration direction. As a whole, the VIV-SL-TENG shows promising potential in developing a fluid kinetic energy harvester. 

## Figures and Tables

**Figure 1 nanomaterials-13-01036-f001:**
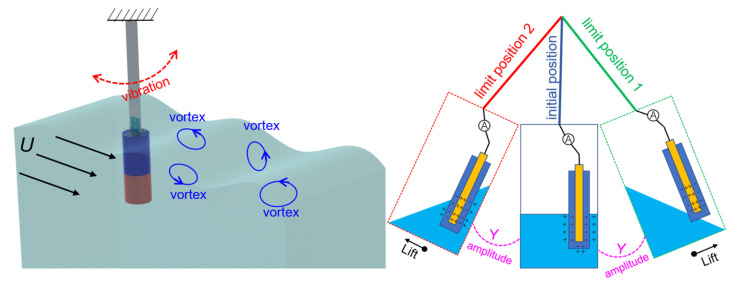
Design concept of VIV-SL-TENG.

**Figure 2 nanomaterials-13-01036-f002:**
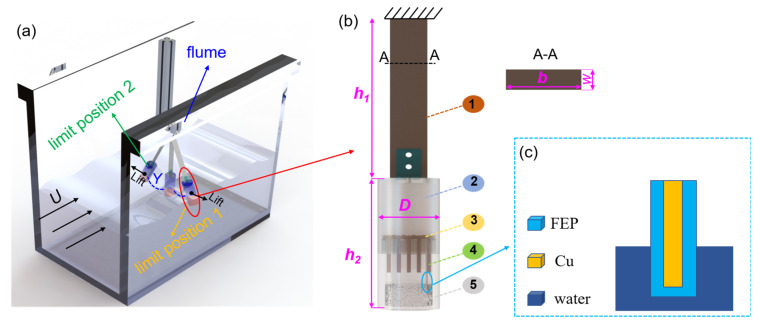
(**a**) TENG system. (**b**) Structural design model of the harvester: (1) cantilever beam, (2) spoiler cylinder, (3) guiding disc, (4) carrier sheet, (5) deionized water. (**c**) Structure of TENG; (**d**) working principle of the SL-TENG.

**Figure 3 nanomaterials-13-01036-f003:**
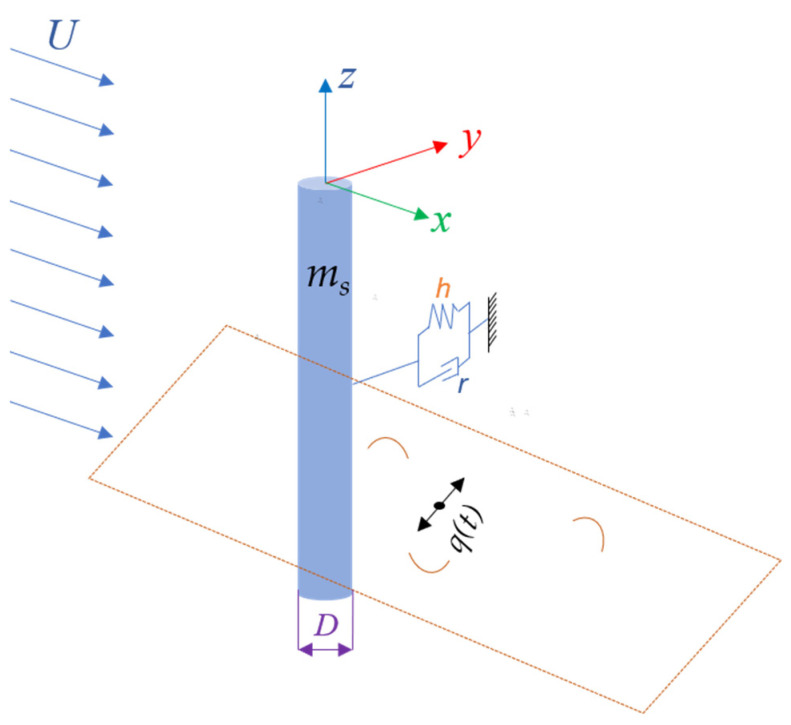
Model of coupled structure and wake oscillators.

**Figure 4 nanomaterials-13-01036-f004:**
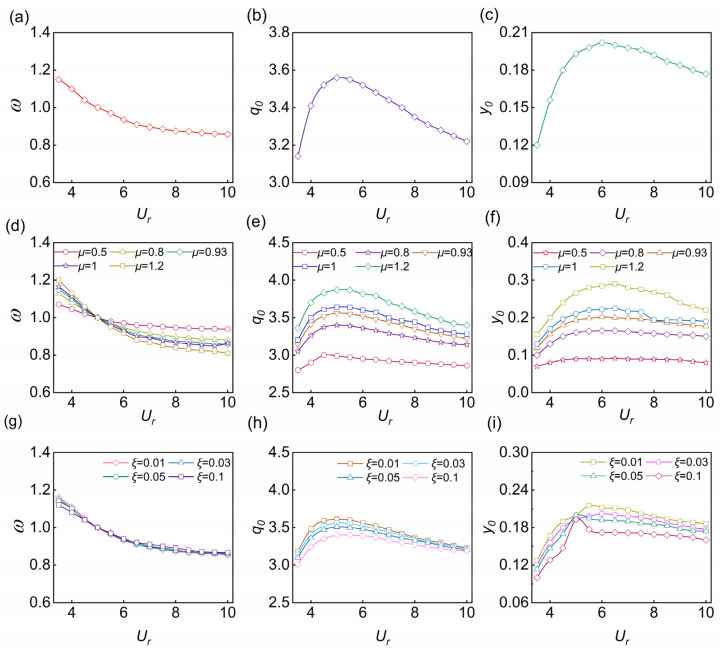
(**a**) Frequency ratio *ω*, (**b**) lift *q*_0_, and (**c**) amplitude *y*_0_ as a function of reduced speed *U_r_*. Influence of mass ratio on (**d**) frequency ratio, (**e**) lift, and (**f**) amplitude. Influence of structural damping on (**g**) frequency ratio, (**h**) lift force, and (**i**) amplitude.

**Figure 5 nanomaterials-13-01036-f005:**
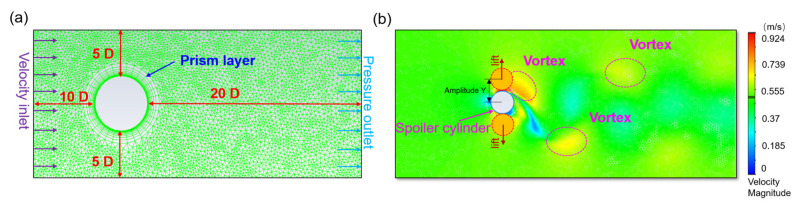
(**a**) Grid of computational domain. (**b**) Scalar diagram of the flow field. (**c**) Lift force, (**d**) displacement, and (**e**) frequency of the cylinder. Effect of flow speed on (**f**) lift force, (**g**) amplitude, and (**h**) frequency. Effect of flow diameter on (**i**) lift force, (**j**) amplitude, and (**k**) frequency.

**Figure 6 nanomaterials-13-01036-f006:**
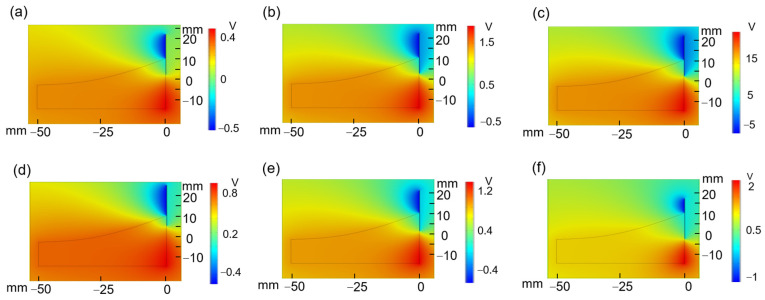
Finite element simulation. Potential distribution when the liquid is (**a**) ethanol, (**b**) deionized water, and (**c**) gallium; and (**d**) potential distribution when the contact area is 5 mm, (**e**) 10 mm, and (**f**) 15 mm.

**Figure 7 nanomaterials-13-01036-f007:**
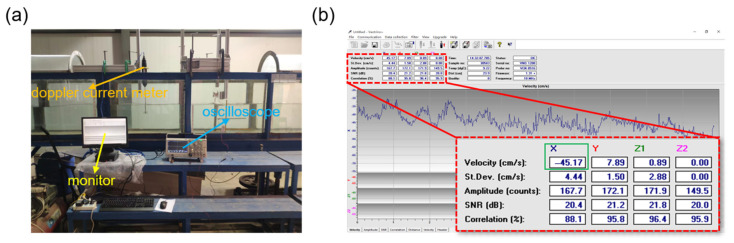
(**a**) Experimental platform. (**b**) Monitor.

**Figure 8 nanomaterials-13-01036-f008:**
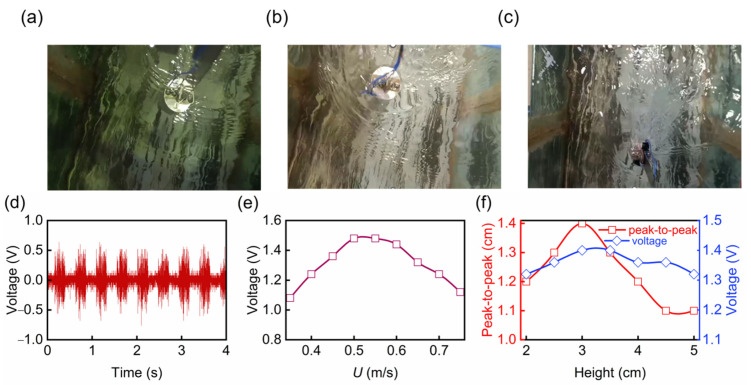
Snapshots of the energy harvester with the cylinder: (**a**) 0.4 m/s, (**b**) 0.5 m/s, (**c**) 0.6 m/s. (**d**) The output voltage of the generator under 0.5 m/s. (**e**) The output voltage of the generator under different flow velocities. (**f**) The output voltage and peak-to-peak of the generator under different heights.

**Figure 9 nanomaterials-13-01036-f009:**
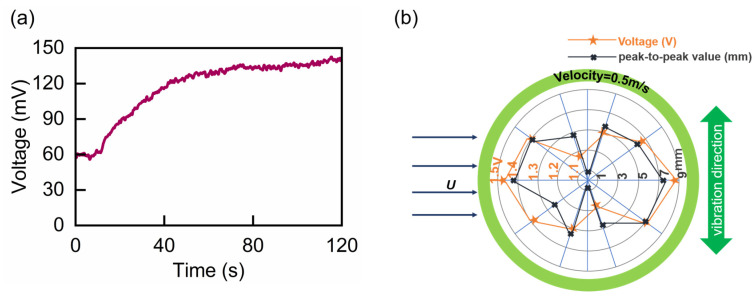
(**a**) Charging performance of the capacitors 0.5 m/s. (**b**) The output voltage and peak-to-peak of the generator under different deflection angles.

**Table 1 nanomaterials-13-01036-t001:** Physical model and structure parameters.

Description	Symbol	Value
Diameter of the cylinder	*D* (mm)	50
Natural frequency	*f* (Hz)	2.017
Strouhal number	$St=\frac{f_{s}\cdot D}{U}$	0.2
Flow speed	*U* (m/s)	0.5
Width of the vibration beam	*b* (mm)	30
Length of the spoiler cylinder	*h*_2_ (mm)	130
Length of the vibration beam	*h*_1_ (mm)	290
Thickness of the vibration beam	*w* (mm)	1

## Data Availability

The data that support the findings of this study is available from the corresponding author upon reasonable request.

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
