# Peer review of "Solid-Liquid Triboelectric Nanogenerator Based on Vortex-Induced Resonance"

_nanomaterials, 2023, doi:10.3390/nano13061036_

Round 1

Reviewer 1 Report

The authors fabricated a solid-liquid triboelectric nanogenerator based on vortex-induced resonance (VIV-SL-TENG). The authors need to address the following before considering their manuscript for publication. 

1. "Traditional solid-solid triboelectric nanogenerators (TENG) are not wear-resistant in corrosive environments", what the authors mean by traditional TENGs? Is TENG technology even traditional?

2. Resolution of the figures should be increased.

3. Mechanism should be elaborated.

4. Output of the device is very low and one can't use it for practical applications. 

5. The authors should have tried for improving the device's output. 

Author Response

Please kindly find the attached file for point-to-point replies to the comments and suggestions.

Reviewer 2 Report

In my opinion, the manuscript "Solid-liquid triboelectric nanogenerator based on vortex-in-duced resonance" is controversial. On the one hand, the relevance and novelty of the manuscript is obvious, since it is devoted to the generation of "blue" energy and the reduction of carbon emissions, and one can also assume the reliability of the results. On the other hand, the manuscript does not sufficiently disclose the component of nanomaterials, nanoscience and nanotechnology to match the name "nanogenerator" and the publication "Nanomaterials". The following fixes are needed: Could you be so kind as to reveal in more detail the prefix "nano-" in your device, perhaps in the preparation of materials or in the technologies used.

Author Response

(The authors gave the same response as above.)

Reviewer 3 Report

see attached file

Author Response

(The authors gave the same response as above.)

Round 2

Reviewer 3 Report

Due to improved figures and text, the revised version is much better to read and to understand.

I recommend publication of the revised version.